# In Situ XANES Studies on Extracted Copper from Scrap Cu/ITO Thin Film in an Ionic Liquid Containing Iodine/Iodide

**DOI:** 10.3390/molecules27061771

**Published:** 2022-03-08

**Authors:** Hsin-Liang Huang, Yu Jhe Wei

**Affiliations:** Department of Safety, Health and Environmental Engineering, National United University, Miaoli 36063, Taiwan; e243158@gmail.com

**Keywords:** ITO thin film, copper, ionic liquid, recovery, extraction, XANES, EXAFS

## Abstract

Copper is coated on indium-tin-oxide (ITO) thin film to improve its electrical resistivity. In order to recycle the scrap Cu/ITO thin film, an ionic liquid (1-butyl-3-methyl imidazolium hexafluorophosphate ([C_4_mim][PF_6_])) containing iodine/iodide (IL-I) was used to extract copper at 303, 343, 413, 374, and 543 K. The extraction efficiency of copper from the scrap Cu/ITO thin film was >99% with IL-I. Using XRD, crystal In_2_O_3_ was found on the regenerated ITO thin film which had a resistivity similar to that of unused ITO thin film. Using X-ray absorption near edge structural (XANES) spectroscopy, at least two paths for the extraction of copper from the Cu/ITO thin film into IL-I were identified. Path I: Copper is stripped from the scrap Cu/ITO thin film and then interacts with I_3_^−^ in the IL-I to form nano CuI. The nano CuI further interacts with I^−^. Path II: Copper interacts with I_3_^−^ on the surface of the Cu/ITO thin film to form nano CuI. The nano CuI is further stripped into the IL-I to interact with I^−^. During extraction, the nanoparticle size could be increased in the IL-I by conglomeration due to fewer coordinating anions and decrease in the viscosity of IL-I at high temperatures. Furthermore, nanoparticle growth was affected by [PF_6_]^−^ of the IL-I determined via ^31^P NMR.

## 1. Introduction

Indium-tin-oxide (ITO) thin film is composed of In_2_O_3_ and SnO_2_ and applied on transparent electrodes for touch screens, EMI shielding, storage devices, and photovoltaic solar cells [1,2,3,4]. ITO thin film has high conductivity, and good transparency and stability. In order to improve the optoelectrical properties and appropriable optical band gaps, the copper is doped onto the ITO thin film [5,6]. The electrical resistivity and band gap of Cu/ITO thin film are decreased by doping copper nanoparticles, which can be applied onto flexible optoelectronic devices with high substrate temperatures, etc. [7]. Demand for ITO thin film has increased as usage and applications have increased. The price of indium was ~180 USD/kg in 2021 [8]. Moreover, indium is an uncommon element and is expected to be depleted. Furthermore, the price of tin increased significantly from 15,000 to 43,000 USD/ton in 2021 [9]. The prices of indium and tin will continue to increase due to increase in demand. Water, dilute aqueous base or acid is used to dissolve indium and tin from ITO thin film and the metals are then separated from the solutions [10]. Polyethyleneimine/sodium silicate and polyethyleneimine/κ-carrageenan composite are novel materials for the removal of metal ions from waste [11,12]. Appropriate technologies should be developed to recycle intact ITO thin film.

Ionic liquids have unique properties, such as low vapor pressure, large liquid temperature ranges, and wide electrochemical windows. Ionic liquids are used in synthesis, electrochemistry, catalysis, carbon capture, and separation [13,14,15,16,17]. In(III) can be extracted by ionic liquids, such as *n*-hexyl-trimethyl ammonium bis(trifluoromethyl-sulfonyl)amide and A327H Cl (by reaction of tertiary amine A327 and HCl) [18]. The ionic liquid, methyltrioctyl/decylammonium bis 2,4,4-(trimethylpentyl)phosphinate extracted Fe(II) and Zn(II) selectively from a solution containing Cu(II), Fe(II), Mn(II), and Zn(II) [19]. Another ionic liquid, tetrahexylammonium dicyanamide, has exhibited selective extraction for Cu(II) [20]. Moreover, protic ionic liquids without chelating agents have also achieved highly selective extraction of copper from water [21]. Although metal ions can be extracted into ionic liquids, metals must dissolve in solutions [22]. We used 1-butyl-3-methyl imidazolium hexafluorophosphate ([C_4_mim][PF_6_]) containing iodine/iodide to extract copper from Cu/ITO. The resistivity and crystal phase of the regenerated ITO thin film was similar to that of unused ITO thin film [22]. In this study, [C_4_mim][PF_6_], containing iodine/iodide (IL-I), was used to extract Cu from Cu/ITO. The main objective of the present study was to investigate the chemical structure of extracted copper from discarded Cu/ITO thin films in IL-I at 303–543 K by in situ X-ray absorption near edge structural (XANES) spectroscopy and extended X-ray absorption fine structural (EXAFS) spectroscopy. The extraction efficiency could be improved by increasing the temperature during extraction. The patterns of nanoparticles in IL-I during the extraction also changed. We identified the changes in nanoparticles which were suitable for reuse. The molecular-scale data were useful in revealing the process of copper and ITO separation from the scrap Cu/ITO thin film.

## 2. Results and Discussion

About 99% of the copper on the surface of the scrap Cu/ITO thin film was extracted into the IL-I at 303–543 K within 30 min (see Figure 1). At 473 K, the extraction efficiency was approximately 99%. As shown in Figure 2, the copper was barely observed on the regenerated ITO thin film after extraction with the IL-I from 303 K to 543 K within 30 min by XRD. Moreover, the crystal In_2_O_3_ was still maintained on the regenerated ITO thin film. The resistivity of the regenerated ITO thin film was 1.99 Ω^.^sq after extraction using a four-point probe. Note that the resistivity of unused ITO thin film was 1.98 Ω^.^sq. The Cu/ITO thin films can be regenerated by extracting copper species with the IL-I at 303–543 K.

The in situ Cu K-edge XANES spectra of the extracted Cu in the IL-I are shown in Figure 3. In the pre-edge XANES, the absorption peak at 8981–8984 eV had dipole-allowed 1s-to-4p_xy_ electron transition, meaning the Cu(I) could be observed in the IL-I (see Figure 3). CuI_2_^−^ (56%), CuI (39%), and Cu (5%) were the main species in the IL-I at 303 K (see Figure 2. As the temperature continued to increase from 303 to 543 K, the concentrations of CuI_2_^−^ increased from 56% to 97% (see Figure 3). We also found that the concentrations of CuI decreased in the IL-I at 303–413 K because CuI interacted with the iodide in IL-I (see Figure 3a–c). The copper species on scrap Cu/ITO thin film extracted into the IL-I may undergo the following reactions: I_2_ + I^−^ → I_3_^−^, 2Cu + I_3_^−^ → 2CuI + I^−^, and 2I^−^ + 2CuI → 2CuI_2_^−^ [22]. Moreover, CuO was formed by oxidation of Cu with dissolved oxygen in the IL-I, as shown in Figure 3b–e. Therefore, the Cu was stripped from the Cu/ITO thin film into the IL-I and then interacted with I_3_^−^ to form CuI during extraction at 303–543 K. Furthermore, CuI_2_^−^ was formed by the interaction of 2I^−^ and CuI.

The copper K-edge least-square fitted XANES spectra of the scrap Cu/ITO thin film during extraction at 303, 343, 413, 473, and 543 K are shown in Figure 4. The absorption peak at 8981–8941 eV can be attributed to the 1s-to-4p_xy_ electron transition of Cu(I). CuI was the main species on the Cu/ITO thin film at 303–543 K. Therefore, I_3_^−^ interacted with Cu and then formed CuI on the surface of the scrap Cu/ITO thin film. Moreover, CuI stripped from the surface of scrap Cu/ITO thin film into the IL-I can interact with I^−^ to form CuI_2_^−^. In XRD, CuI on the scrap Cu/ITO thin film was observed during extraction with IL-I at 303, 343, 413, 473, and 543 K, respectively (see Figure 5). The crystalline sizes of CuI calculated by the Debye–Scherrer formula were 35, 49, 64, 67, and 65 nm on the scrap Cu/ITO thin film after extraction with the IL-I at 303, 343, 413, 473, and 543 K, respectively. The increased crystalline sizes meant more long-range order at 303, 343, 413, and 473 K. At 543 K, the extraction efficiency was approximately 99%. The residual copper on the Cu/ITO thin film still reacted with I_3_^−^ and I^−^. Therefore, the crystalline size was decreased at 543 K. The increased crystalline sizes were limited at 413–543 K because crystals were grown at the interface of the scrap Cu/ITO thin film and IL-I; therefore, the crystal growth was limited by the IL-I.

In order to understand the interaction between copper species and IL-I, the ^1^H and ^31^P NMR of IL and IL-I during extraction were measured. In Figure 6, the protons of IL-I were little perturbed during extraction because IL is a noncoordinating solvent. We found that protons of the IL were slightly disturbed by I_2_, I^−^, I_3_^−^, and CuI_2_^−^ by ^31^P NMR spectra (see Figure 7a–c). However, the intensities at δ −144.8 decreased in Figure 7d–f because Figure 3a–c shows that 44%, 17%, and 8% of particles can affect the [PF_6_]^−^ of the IL-I at 298, 343, and 413 K, respectively. The particle sizes of CuI are shown in Figure 8. The size distributions of particles were obtained by fitting the in situ synchrotron small-angle X-ray scattering (SAXS) patterns with the Schultz distribution function. The particle sizes of copper species were 32, 50, 64, 70, and 62 nm on the scrap Cu/ITO thin film during extraction with IL-I at 303, 343, 413, 473, and 543 K, respectively. The particles can be conglomerated due to the less coordinating anions and decreasing the viscosity of IL at high temperature. Furthermore, the particle sizes were around 62–70 nm as the temperature rose above 413 K. Moreover, electron-deficiency was observed on the surface of the metal nanoparticles. An electric double layer is formed on the surface of the nanoparticles in ionic liquids. The first exterior layer of the metal nanoparticles is comprised of the anions of ionic liquids, while the cations of ionic liquids comprise the second coordination layer [23]. Therefore, the formation of nanoparticles interacted with the [PF_6_]^−^ of IL-I in Figure 8b–d. CuI nanoparticle size on the scrap Cu/ITO depended on the [PF_6_]^−^ of IL-I. In Figure 8e,f, the concentrations of particles were too low for interaction of the particles and the [PF_6_]^−^ of IL-I to occur.

The EXAFS spectra of copper in the IL-I and on the Cu/ITO during extraction were also recorded and analyzed in the k range of 3.5–11.5 Å^−1^. An over 95% reliability of the Fourier transformed EXAFS data fitting for copper was obtained. In all the EXAFS data analyses, the Debye–Waller factors (Δσ^2^) were less than 0.01 Å^2^ (Table 1 and Table 2). Table 1 shows that the bond distance of Cu-I was 2.55 Å with CNs (coordinated numbers) of 1.7–1.9 for CuI_2_^−^ in the IL-I at 303–543 K. However, Table 2 shows that the bond distance of Cu-I for CuI on the Cu/ITO was 2.56–2.57 Å with CNs of 3.1–3.4 at 303–543 K.

## 3. Materials and Methods

The synthesis process of [C_4_mim][PF_6_] has been described previously [24]. A quantity of 5 mL of NaI (0.2 M) (99.5%, Merck, Darmstad, Germany) and I_2_ (0.04 M) (99.8–100.5%, Merck, Darmstad, Germany) were added to 1 g of [C_4_mim][PF_6_] to form IL-I. The thickness of copper coated onto the surface of ITO thin films was approximately 380–400 nm. The Cu/ITO was collected from touch panels in Taiwan. It was estimated that 0.03 g of copper on the scrap Cu/ITO thin films was extracted into one g of IL-I from 303 K to 543 K within 30 min. The spectra of samples were collected at 303, 343, 413, 473, and 543 K during extraction. The samples after IL-I extraction were digested and the copper concentrations were then measured by AA (Hitachi Z-5000, Hitachi Instruments Co., Tokyo, Japan). The extraction efficiencies of copper from the Cu/ITO to IL-I were estimated by the following equation:Extraction efficiency (%) = *C*_2_/*C*_1_ × 100%
where *C*_1_ (mg/g) is the initial concentration of copper in Cu/ITO and *C*_2_ (mg/g) is the concentration of extracted copper in IL-I.

The regenerated ITO thin film was determined by XRD (D8 Advance, Bruker, Billerica, MA, USA) with Cu Kα (1.542 A°) radiation. Samples were scanned from 20° to 60° (2*θ*). The resistivity of the thin films was measured with a four-point probe (Quatek Model: QT-50, Shanghai, China). ^1^H and ^31^P NMR (nuclear magnetic resonance) spectra of the samples were also determined with a Bruker Advanced 300 spectrometer (Bruker, Billerica, MA, USA) with tetramethyl silane (TSM) as an internal standard (acquisition time = 1.373 s, actual pulse repetition time = 2 s, number of scans = 32 and excitation pulse angle = 30°).

The in situ EXAFS and XANES spectra of samples were recorded using a Wiggler BL17C at the Taiwan National Synchrotron Radiation Research Center in an electron storage ring with energy of 1.5 GeV and a current of 80–200 mA. An Si(111) double-crystal monochromator was used for energy selection with an energy resolution (ΔE/E) of 1.9 × 10^−4^ (eV/eV). The photon energy was calibrated using a copper foil absorption edge at 8979 eV. The standard deviation calculated from the average spectra was used to estimate the statistical noise and error associated with each structural parameter. The absorption edge was determined at the half-height (precisely determined by its derivative) of the XANES spectrum after pre-edge baseline subtraction and normalization to the maximum above-edge intensity. The spectra of samples and model compounds, such as CuO (≥99.0%, Merck, Darmstad, Germany), Cu (99.7%, Merck, Darmstad, Germany), CuI (99%, Sigma-Aldrich, St. Louis, MO, USA), and CuI_2_^−^, were also collected on the beam line. An amount of 0.5 g of CuI was dissolved in 5 mL of NaI (1 M) (99.5%, Merck) to prepare the CuI_2_^−^. The EXAFS data were analyzed using UWXAFS 3.0 and FEFF 8.0 simulation programs [25].

The in situ SAXS spectra were recorded using a 23A SWAXS at the Taiwan National Synchrotron Radiation Research Center in an electron storage ring with energy of 1.5 GeV and a current of 80–200 mA. A Mo/B_4_C-multilayer monochromator was used for energy selection with an energy resolution (ΔE/E) of 1 × 10^−2^ (eV/eV) under the energy of 14 keV. The thickness of the samples was 1.9 mm, which were heated from 303–543 K at a heating rate of 8 K/min in N_2_ gas with 100 mL/min [26,27]. The particle sizes of samples were analyzed by TableCurve software (Jandel Scientific, San Rafael, CA, USA).

## 4. Conclusions

Approximately 99% of copper on scrap Cu/ITO was extracted with the IL-I at 303–473 K. Using XRD, crystal In_2_O_3_ was still observed on the regenerated ITO thin film after extraction. The resistivity of the regenerated ITO thin film was 1.99 Ω^·^sq. Using XANES, the copper on the scrap Cu/ITO was shown to have two extraction pathways into the IL-I. One path was copper stripped from Cu/ITO and then interacting with I_3_^−^ in the IL-I to form nano CuI, which further interacted with I^−^ to form CuI_2_^−^. The other path was copper interacting with I_3_^−^ on the surface of Cu/ITO to form nano CuI; nano CuI was then stripped into IL-I to interact with I^−^. At high temperatures, the nanoparticles collided and aggregated to increase particle sizes due to fewer coordinating anions and decrease in the viscosity of IL-I. Moreover, ^31^P NMR showed that the [PF_6_]^−^ of IL-I could affect the growth of nanoparticles.

## Figures and Tables

**Figure 1 molecules-27-01771-f001:**
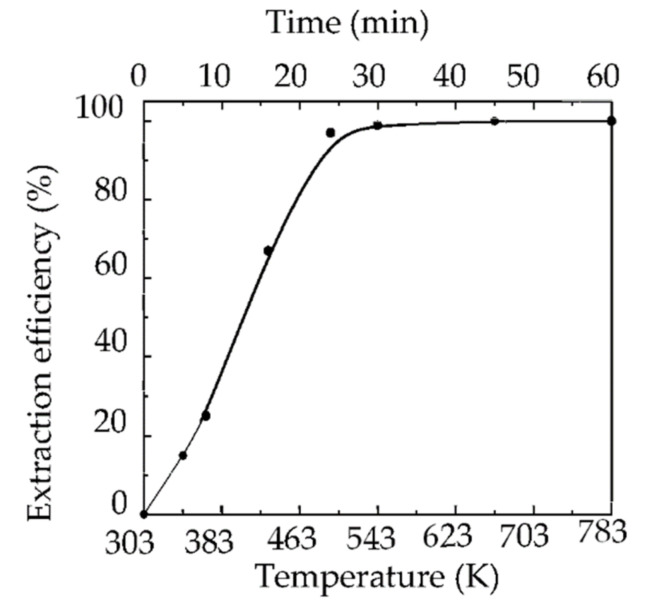
Temperature course for extraction of copper from the scrap Cu/ITO thin film with IL-I.

**Figure 2 molecules-27-01771-f002:**
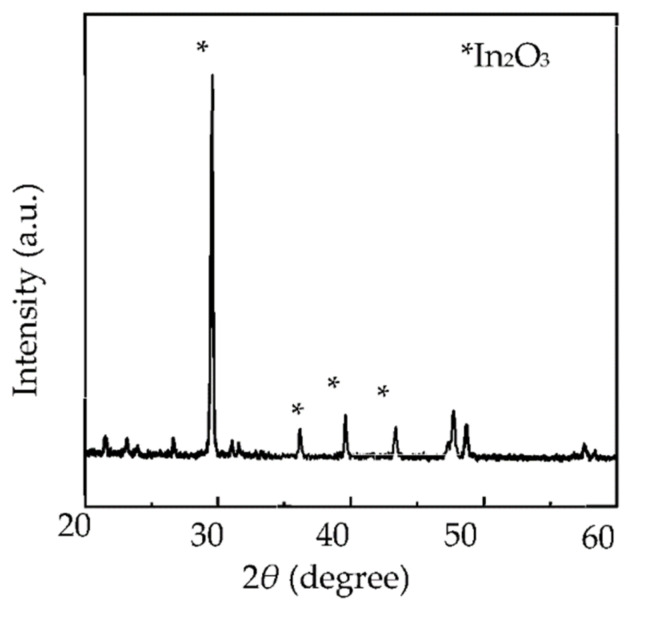
XRD patterns of the regenerated ITO thin film after extraction of copper species with IL-I at 303–543 K.

**Figure 3 molecules-27-01771-f003:**
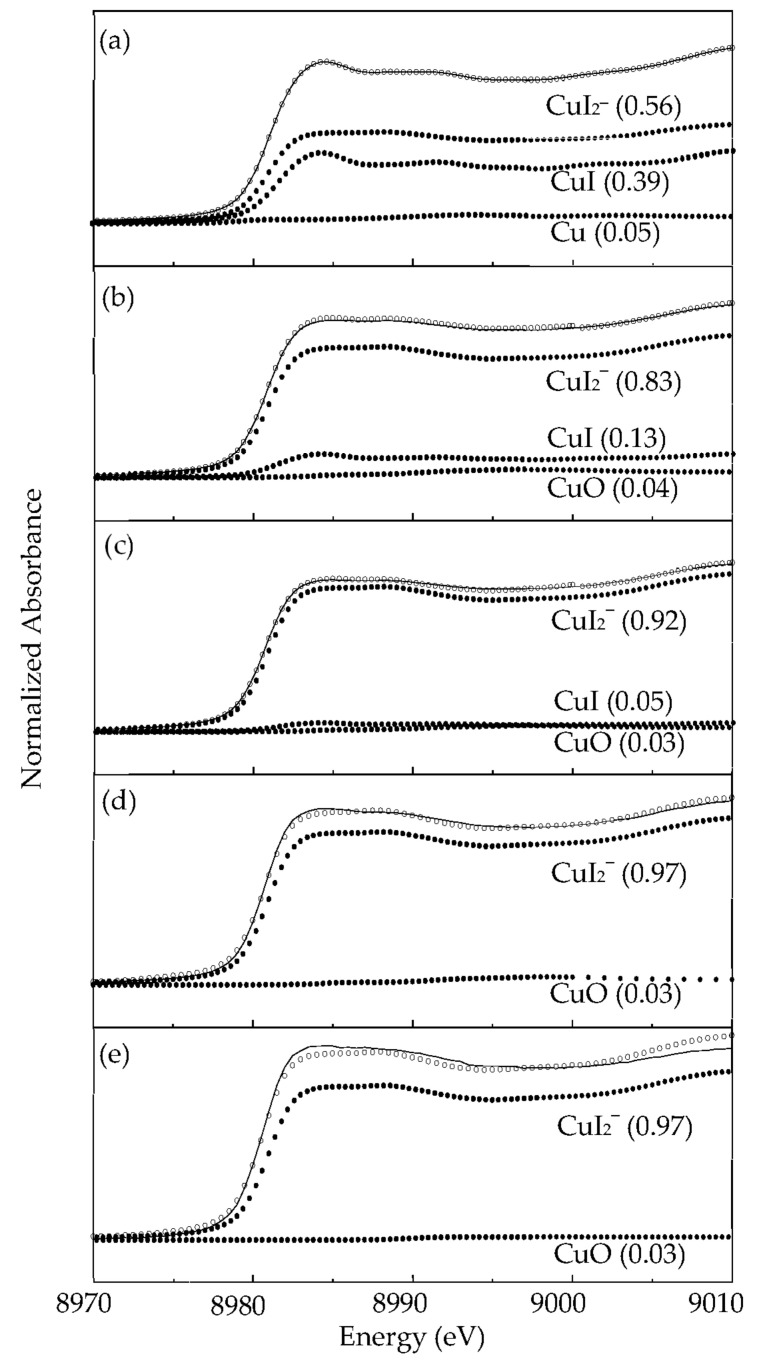
Experimental data (solid line) and the least-squares fit (circles) of in situ XANES spectra of extracted copper in the IL-I at (**a**) 303 (**b**) 343 (**c**) 413 (**d**) 473, and (**e**) 543 K.

**Figure 4 molecules-27-01771-f004:**
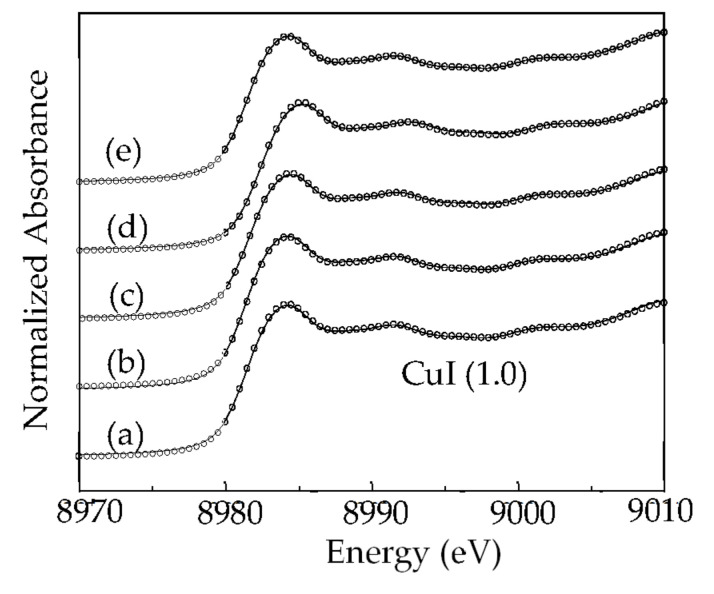
Experimental data (solid line) and the least-squares fit (circles) of XANES spectra of residual copper on the Cu/ITO thin film during extraction with the IL-I at (**a**) 303 (**b**) 343 (**c**) 413 (**d**) 473, and (**e**) 543 K.

**Figure 5 molecules-27-01771-f005:**
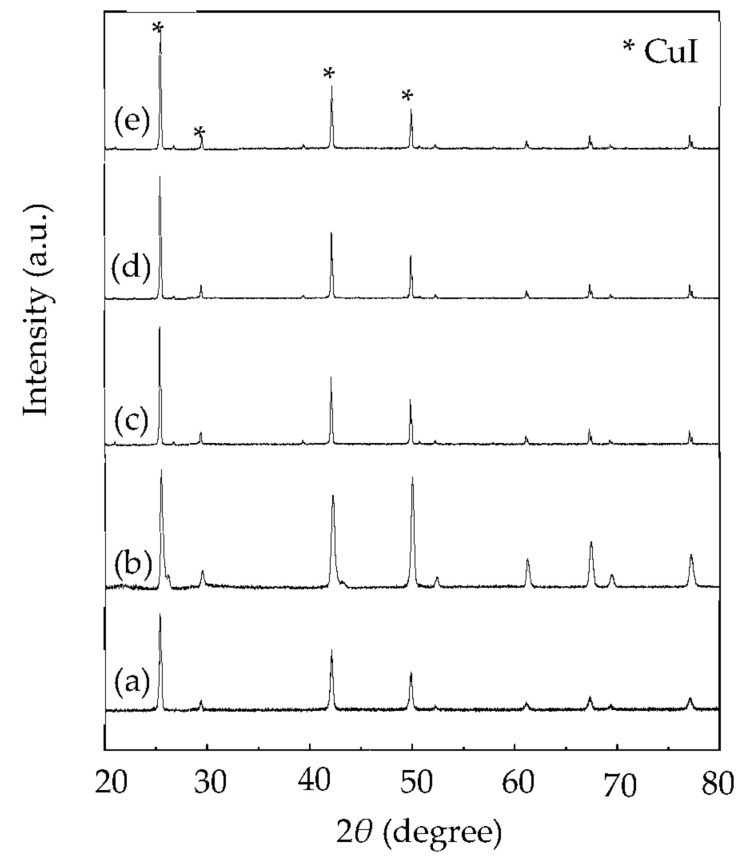
XRD patterns of residual copper on the Cu/ITO thin film during extraction with the IL-I at (**a**) 303, (**b**) 343, (**c**) 413, (**d**) 473, and (**e**) 543 K.

**Figure 6 molecules-27-01771-f006:**
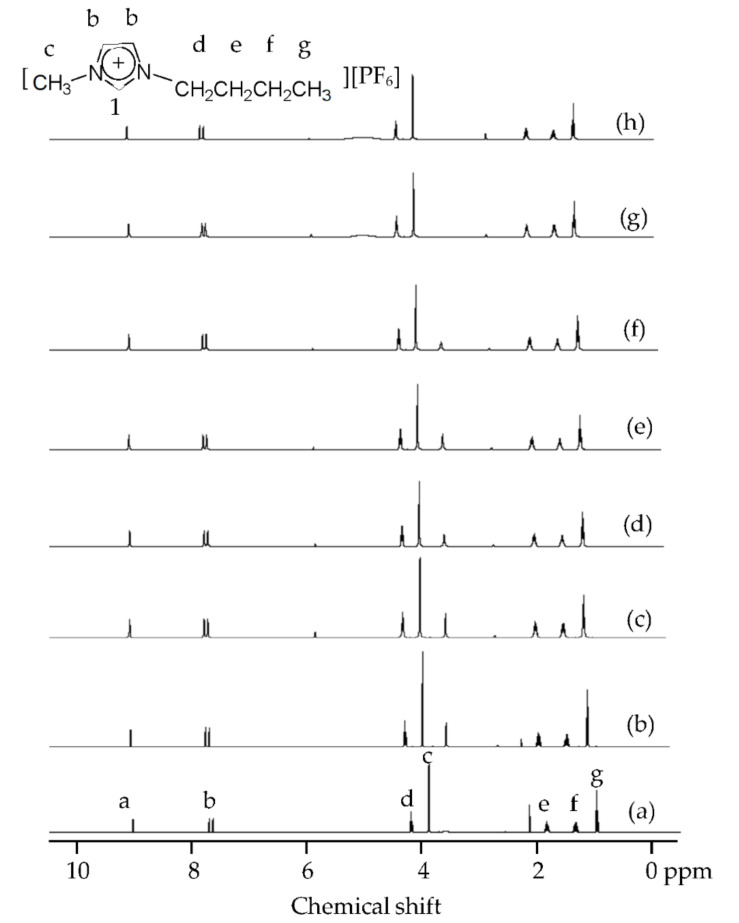
^1^H NMR spectra of the (**a**) IL, (**b**) IL-I, (**c**) IL-I containing CuI_2_^−^ and extracted-copper IL-I at (**d**) 303 (**e**) 343 (**f**) 413 (**g**) 473, and (**h**) 543 K.

**Figure 7 molecules-27-01771-f007:**
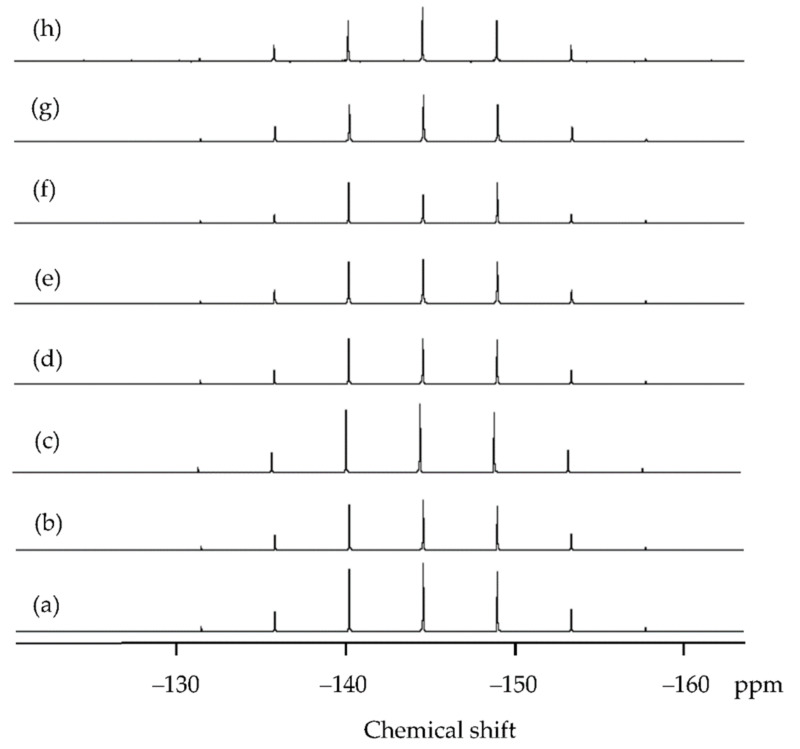
^31^P NMR spectra of the (**a**) IL, (**b**) IL-I, (**c**) IL-I containing CuI_2_^−^ and extracted-copper IL-I at (**d**) 303 (**e**) 343 (**f**) 413 (**g**) 473, and (**h**) 543 K.

**Figure 8 molecules-27-01771-f008:**
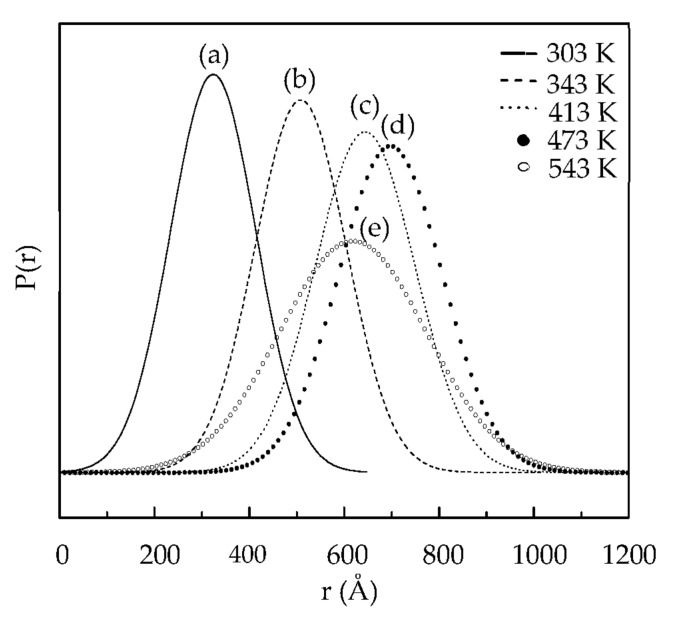
Particle size distributions of copper in the IL-I at (**a**) 303, (**b**) 343, (**c**) 413, (**d**) 473, and (**e**) 543 K with in situ SAXS fitting analysis.

**Table 1 molecules-27-01771-t001:** Structural parameters of extracted copper in the IL-I at 303–543 K.

	Shell	Bond Distance (Å)	Coordination Number	σ^2^ (Å^2^)
Extracted copper in the IL-I				
at 303 K	Cu-I	2.55	1.9	0.007
at 343 K	Cu-I	2.55	1.7	0.008
at 413 K	Cu-I	2.55	1.8	0.008
at 473 K	Cu-I	2.55	1.9	0.008
at 543 K	Cu-I	2.55	1.8	0.008

σ^2^: Debye–Waller factor.

**Table 2 molecules-27-01771-t002:** Speciation of residual copper on the Cu/ITO thin film during extraction with the IL-I at 298–543 K.

	Shell	Bond Distance (Å)	Coordination Number	σ^2^ (Å^2^)
Residual copper on the Cu/ITO thin film				
at 303 K	Cu-I	2.57	3.4	0.01
at 343 K	Cu-I	2.57	3.1	0.01
at 413 K	Cu-I	2.56	3.4	0.01
at 473 K	Cu-I	2.57	3.2	0.01
at 543 K	Cu-I	2.57	3.4	0.01

σ^2^: Debye–Waller factor.

## Data Availability

Not applicable.

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
