# Peer review of "In Situ XANES Studies on Extracted Copper from Scrap Cu/ITO Thin Film in an Ionic Liquid Containing Iodine/Iodide"

_molecules, 2022, doi:10.3390/molecules27061771_

Round 1
Reviewer 1 Report
The main objective of the present work was to investigate the chemical structures of extracted copper from discarded Cu/ITO thin films in IL-I at 303-543 K by in situ X-ray absorption near edge structural (XANES) spectroscopy and extended X-ray absorption fine structural (EXAFS) spectroscopy. This manuscript could be considered for publication in Molecules after a minor revision.
- Please mark (a), (b), (c), (d)and (e) in Figure 8.
- Figures2 and 5: Please add the x-axis unit “degree”.
- Please modify the line number “122-140”.
- The particle sizes of copper species were 32, 50, 64, 70, and 62 nm on the scrap Cu/ITO thin film during extraction with IL-I at 303, 343, 413, 473, and 543 K, respectively.
Please analyze the reason why the particle size of copper material first increases and then decreases with the increase of temperature.
- Line 180: “Figures 8(e) and (f)” seem to be replaced by “Figures 7(e) and (f)”.
- Please revise references 20, 22 and 24 according to the author guidelines.
- The relatedreferences are suggested to be cited such as Journal of Materials Science,2022 57, 4221–4238; Journal of Polymers and the Environment, 2022, 30,1001–1011.
Author Response
Reviewer #1
The main objective of the present work was to investigate the chemical structures of extracted copper from discarded Cu/ITO thin films in IL-I at 303-543 K by in situ X-ray absorption near edge structural (XANES) spectroscopy and extended X-ray absorption fine structural (EXAFS) spectroscopy. This manuscript could be considered for publication in Molecules after a minor revision.
- Please mark (a), (b), (c), (d) and (e) in Figure 8.
Response:
The (a), (b), (c), (d) and (e) have marked in Figure 8.
- Figures 2 and 5: Please add the x-axis unit “degree”.
Response:
The “degree” has added on the x-axis in Figures 2 and 5.
- Please modify the line number “122-140”.
Response:
The line number “122-140” has been modified.
- The particle sizes of copper species were 32, 50, 64, 70, and 62 nm on the scrap Cu/ITO thin film during extraction with IL-I at 303, 343, 413, 473, and 543 K, respectively.
Please analyze the reason why the particle size of copper material first increases and then decreases with the increase of temperature.
Response:
The reasons have been written in the Results and Discussion.
……The crystalline sizes of CuI calculated by Debye-Scherrer formula were 35, 49, 64, 67, and 65 nm on the scrap Cu/ITO thin film after extraction with the IL-I at 303, 343, 413, 473, and 543 K, respectively. The increased crystalline sizes meant more long-range order at 303, 343, 413, and 473 K. At 543 K, the extraction efficiency was approximately 99%. The residual copper on the Cu/ITO thin film still reacted with I3- and I-. Therefore, the crystalline size was decreased at 543 K. The increased crystalline sizes were limited at 413-543 K because crystals were grown at the interface of the scrap Cu/ITO thin film and IL-I; therefore, the crystal growth should be limited by the IL-I.
- Line 180: “Figures 8(e) and (f)” seem to be replaced by “Figures 7(e) and (f)”.
Response:
The “Figures 8(e) and (f)” at line 180 has been revised.
- Please revise references 20, 22 and 24 according to the author guidelines.
Response:
The references 20, 22 and 24 have been revised.
- The related references are suggested to be cited such as Journal of Materials Science, 2022 57, 4221–4238; Journal of Polymers and the Environment, 2022, 30,1001–1011.
Response:
As suggested, No.11 and 12 of references from Journal of Materials Science and Journal of Polymers and the Environment in the Introduction were cited.

Reviewer 2 Report
The submitted manuscript reports on the extraction of copper from scrap Cu/ITO thin films using bmimPF6 containing iodine/iodide. The authors have already published a number of papers on a similar topic, and this manuscript adds insights on the chemical structure of extracted copper, in the process studied previously, in 2016 (Chem. Phys. Lett., 2016, 652, 195).The manuscript is well organised. Objectives are clearly stated and an appropriate, brief background is provided. The methods are well described and the materials thoroughly characterised. I consider this contribution interesting, and I recommend its publication after minor changes.
- more clear distinction between what was done in the previous and current work is needed
- the abstract is messy, the authors’ conclusions section makes better summary than the text in the abstract, please rewrites
- uggestion: [1-4] instead of [1,2,3,4]
- A327H Cl should be explained – it’s not a common IL and needs some explanation, even if it’s “mixture of tri-octil and tri-decyl amines…”; also, bis(2,4,4-trimethylpentyl)phosphinate is missing “e” at the end
- there seems to be some problem with the resistivity unit
- I’d like to see a comment on a recyclability of the IL; is it possible? was there any attempt to separate the copper from the IL?
- what about the stability of this IL at higher temperatures? data at 780K?
